# Hearts, Data, and Artificial Intelligence Wizardry: From Imitation to Innovation in Cardiovascular Care

**DOI:** 10.3390/biomedicines13051019

**Published:** 2025-04-23

**Authors:** Panteleimon Pantelidis, Polychronis Dilaveris, Samuel Ruipérez-Campillo, Athina Goliopoulou, Alexios Giannakodimos, Panagiotis Theofilis, Raffaele De Lucia, Ourania Katsarou, Konstantinos Zisimos, Konstantinos Kalogeras, Evangelos Oikonomou, Gerasimos Siasos

**Affiliations:** 13rd Department of Cardiology, National and Kapodistrian University of Athens, 11527 Athens, Greece; agoliopoulou@gmail.com (A.G.); alexisgiannak@hotmail.com (A.G.); raniakatsarou@yahoo.gr (O.K.); zisimoskostas@gmail.com (K.Z.); kalogerask@yahoo.gr (K.K.); boikono@med.uoa.gr (E.O.); gsiasos@med.uoa.gr (G.S.); 2Department of Computer and Systems Sciences, Stockholm University, 16455 Stockholm, Sweden; 31st Department of Cardiology, National and Kapodistrian University of Athens, 11527 Athens, Greece; hrodil1@yahoo.com (P.D.); panos.theofilis@hotmail.com (P.T.); 4Department of Computer Science, ETH Zurich, 8092 Zurich, Switzerland; samuel.ruiperezcampillo@inf.ethz.ch; 5Department of Medicine, Stanford University, Stanford, CA 94305, USA; 62nd Division of Cardiology, Cardiac Thoracic and Vascular Department, Azienda Ospedaliero Universitaria Pisana, 56124 Pisa, Italy; r.delucia.md@gmail.com

**Keywords:** cardiology, cardiovascular medicine, artificial intelligence, deep learning, machine learning, cardiovascular imaging, electrocardiogram, multi-modal data, omics

## Abstract

Artificial intelligence (AI) is transforming cardiovascular medicine by enabling the analysis of high-dimensional biomedical data with unprecedented precision. Initially employed to automate human tasks such as electrocardiogram (ECG) interpretation and imaging segmentation, AI’s true potential lies in uncovering hidden disease data patterns, predicting long-term cardiovascular risk, and personalizing treatments. Unlike human cognition, which excels in certain tasks but is limited by memory and processing constraints, AI integrates multimodal data sources—including ECG, echocardiography, cardiac magnetic resonance (CMR) imaging, genomics, and wearable sensor data—to generate novel clinical insights. AI models have demonstrated remarkable success in early dis-ease detection, such as predicting heart failure from standard ECGs before symptom on-set, distinguishing genetic cardiomyopathies, and forecasting arrhythmic events. However, several challenges persist, including AI’s lack of contextual understanding in most of these tasks, its “black-box” nature, and biases in training datasets that may contribute to disparities in healthcare delivery. Ethical considerations and regulatory frameworks are evolving, with governing bodies establishing guidelines for AI-driven medical applications. To fully harness the potential of AI, interdisciplinary collaboration among clinicians, data scientists, and engineers is essential, alongside open science initiatives to promote data accessibility and reproducibility. Future AI models must go beyond task automation, focusing instead on augmenting human expertise to enable proactive, precision-driven cardiovascular care. By embracing AI’s computational strengths while addressing its limitations, cardiology is poised to enter an era of transformative innovation beyond traditional diagnostic and therapeutic paradigms.

## 1. Introduction

Artificial Intelligence (AI) has emerged as a game-changer in cardiovascular medicine, pushing the boundaries of what was once thought possible. Over the past decade, rapid advances in computational power and algorithmic sophistication have facilitated the transition from traditional machine learning (ML) techniques to deep learning (DL) models. These models have significantly enhanced diagnostic precision, prognosis assessment, treatment guidance, and risk stratification in cardiovascular care [1].

Despite AI’s potential, its current applications in cardiology and vascular diseases remain largely focused on automating human tasks, such as detecting arrhythmias in electrocardiograms (ECGs) or segmenting cardiac and vascular structures in imaging. While these developments might improve workflow efficiency, they primarily emphasize automation rather than true innovation in leveraging AI for novel discoveries and clinical breakthroughs [2,3]. Human cognition excels in pattern recognition and the application of experiential knowledge but remains inherently limited in tasks requiring large-scale data integration, complex temporal analysis, and parallel processing [4]. AI, on the other hand, is well suited for such challenges, as it can be trained on high-dimensional datasets from multiple sources—including ECG, echocardiography, cardiac magnetic resonance imaging (CMR), genomics, and clinical and wearable sensor data—to extract clinically relevant insights that may not be immediately apparent through conventional analysis (Figure 1). By shifting from task replication to augmentation and discovery, AI holds the potential to advance precision medicine in cardiology, enabling more accurate diagnosis, refined risk stratification, and personalized treatment strategies [5].

This work combines a review of the evolving role of AI in cardiovascular medicine, with a perspective-driven narrative and a forward-looking discussion on its transformative potential and future directions in clinical practice. It synthesizes evidence from recent studies, highlights AI’s strengths, and discusses key challenges in integrating AI-driven tools into routine clinical practice.

## 2. Human Cognition vs. Artificial Intelligence

### 2.1. Literature Search Strategy

We performed a structured literature search across PubMed, Scopus, and IEEE Xplore, covering publications from inception through March 2025. Our aim was to identify high-impact and clinically relevant studies on artificial intelligence in cardiovascular medicine. Search terms included combinations of the following keywords: (“artificial intelligence” or “machine learning” or “deep learning” or “neural networks”) and (“cardiology” or “cardiovascular” or “ECG” or “electrocardiogram” or “echocardiography” or “CMR” or “cardiac imaging” or “arrhythmia” or “heart failure” or “precision medicine” or “multi-omics”).

We included original research articles that focused on the development or implementation of AI methods for diagnosis, prognosis, risk stratification, or treatment of cardiovascular diseases, particularly those involving novel paradigms beyond traditional clinical tasks. We excluded non-English publications, non-original research (e.g., reviews or editorials), and studies unrelated to cardiovascular applications.

We also manually screened the references of the selected papers, through “snowballing” [6], to identify additional relevant studies not captured through database queries.

### 2.2. Human Expertise in Context

The way clinicians and AI process information is fundamentally distinct [7]. Clinicians integrate a combination of logical reasoning, experiential learning, subconscious pattern recognition, and contextual reasoning. When formulating a diagnosis, they synthesize multiple sources of information, including patient history, physical examination findings, laboratory results, imaging studies, and even non-verbal cues such as stress levels, lifestyle factors, or subtle hesitations in speech [8]. This ability to integrate complex, often unstructured data allows for intuitive, adaptable, and highly personalized decision-making.

In contrast, AI systems operate through mathematical optimization, identifying statistical and non-linear correlations across large datasets to mimic clinical workflows and established diagnostic pathways. However, this imitative approach is inherently limited [2,9]. AI often struggles to generalize to unseen data, often exhibiting poor performance in rare disease presentations or when confronted with patients with uncommon risk factors.

Furthermore, while AI excels in pattern recognition, it lacks the ability to incorporate real-world context, such as a patient’s social determinants of health or emotional state. This limitation reduces its adaptability in clinical settings, where dynamic decision-making is needed. Unlike human clinicians, AI does not possess the ability to question its own conclusions, apply ethical reasoning, or account for unseen variables. These constraints highlight why, despite AI’s capability to process vast amounts of data at unprecedented speeds, its real-world diagnostic accuracy often remains comparable, or sometimes inferior to that of experienced physicians [3,9]. For AI to make meaningful contributions to cardiovascular medicine, its role must evolve beyond naïve mimicry toward complementing human expertise by addressing challenges that are difficult or impossible for human cognition alone to resolve.

### 2.3. AI’s True Strength: Tackling Intractable Problems

Unlike human clinicians, who rely on experiential learning and intuition, AI operates through mathematical reasoning, enabling it to process immense amounts of complex data at speeds far beyond human cognition (Figure 2). AI can analyze thousands of ECGs, echocardiograms, or genomic datasets within seconds, uncovering patterns that even the most skilled cardiologists might overlook [10,11]. Its ability to integrate high-dimensional, multimodal data—including imaging, genomics, biomarkers, and continuous monitoring—facilitates a level of analysis that exceeds traditional diagnostic methods.

Moreover, AI’s consistency eliminates cognitive fatigue, variability, and biases inherent in human decision-making. In high-risk clinical settings, where continuous real-time monitoring is essential, AI can provide uninterrupted assessment, detecting subtle physiological changes that might otherwise go unnoticed [12]. Perhaps AI’s most revolutionary potential lies in its ability to uncover hidden disease markers. Traditional diagnostic approaches rely on well-defined clinical features, such as ST-segment elevations in ECGs [13] or morphological abnormalities in echocardiograms [14]. In contrast, AI has demonstrated the capability to detect preclinical disease states by identifying subtle, latent patterns that would be imperceptible through conventional methods [15].

For instance, AI models have successfully predicted heart failure from ECGs long be-fore the onset of overt symptoms [16] and distinguished genetic cardiomyopathies using echocardiographic [17], ECG [18], and multimodal data. These insights represent a paradigm shift in cardiovascular diagnostics, moving from feature-based assessment to a data-driven approach capable of recognizing novel, previously undetectable disease markers.

## 3. From Data to Innovation: AI’s Transformative Applications

AI is no longer confined to detecting abnormalities; it is now uncovering subtle, pre-clinical markers of disease progression. Its applications extend from precision medicine to real-time monitoring from predictive analytics, reshaping cardiovascular diagnostics and risk assessment (Table 1). Next, we present some of the transformative applications of AI through three lenses: electrocardiography, cardiovascular imaging, and precision medicine and omics.

### 3.1. Electrocardiogram

AI models have demonstrated remarkable performance in extracting predictive signatures from standard 12-lead ECGs, facilitating early detection of left ventricular systolic dysfunction and atrial fibrillation even in sinus rhythm [16,19]. Beyond conventional diagnostics, AI-driven ECG analysis has proven effective in detecting hypertrophic cardiomyopathy [18], dynamically predicting malignant ventricular arrhythmias in patients with an implantable cardioverter-defibrillator (ICD) [20], and it has also succeeded in identifying patients with concealed long QT syndrome and even differentiating specific genetic subtypes [21]. These advancements underscore AI’s ability to recognize complex patterns imperceptible to human experts, often surpassing traditional biomarkers such as QTc measurements.

AI is also revolutionizing cardiovascular outcome prediction, particularly for ventricular arrhythmias and sudden cardiac death. For instance, a recent AI-driven model trained on 24-h ambulatory ECG recordings demonstrated superior performance in stratifying patients at risk for ventricular tachycardia compared to conventional risk assessment tools [22]. Moreover, integrating AI with wearable biosensors has enabled real-time prediction of cardiac events, such as atrial fibrillation and ventricular tachycardia [23]. A recent study introduced a DL model that uses smartwatch-derived R-to-R intervals to predict atrial fibrillation onset up to 30 min in advance [24], while another applied machine learning to Holter ECG recordings, significantly improving short-term arrhythmia risk prediction [25].

### 3.2. Cardiovascular Imaging

In cardiovascular imaging, advanced DL models have made significant strides in detecting complex conditions such as cardiac amyloidosis using multimodal data or even ECG alone [26,27], or improving pipelines by allowing automatic segmentation of cardiac chambers in computed tomography (CT) images [28]. AI-assisted echocardiography has further enhanced diagnostic precision by powering lesion localization and disease phenotyping across multiple cardiac conditions, including atrial septal defects, dilated cardiomyopathy, hypertrophic cardiomyopathy, and prior myocardial infarction [17].

Additionally, DL techniques have been successfully leveraged to predict late gadolinium enhancement (LGE) from echocardiographic data, an achievement that could revolutionize fibrosis detection by providing a widely accessible, ultrasound-based alternative to cardiac magnetic resonance imaging (MRI) [29]. AI models have also demonstrated high accuracy in differentiating ischemic from dilated cardiomyopathy [30], while the recent application of DL in echocardiographic imaging has allowed the genetic profiling of patients with hypertrophic cardiomyopathy [31]. Other modalities like MRI have also been used in multimodal settings for various tasks, including the identification of patients with non-ischemic cardiomyopathy at risk of lethal ventricular arrhythmias [32].

### 3.3. Precision Medicine and Omics

AI is also reshaping treatment strategies, particularly in the realm of personalized medicine. AI-driven phenomapping has been employed for guiding chest pain assessment [33], while multimodal imaging integration has optimized patient selection for ICD [34] or to identify phenotypes for acute and long-term response to atrial fibrillation ablation [35]. Moreover, AI is now increasingly integrated into interventional workflows, guiding precision-driven procedural planning across both cardiology and vascular domains. Fields of application include lesion assessment and stent sizing in percutaneous coronary interventions [36,37] or substrate localization and catheter guidance in electrophysiology procedures [38,39]. In vascular surgery and endovascular interventions, AI is adopted to support procedural planning and risk stratification in peripheral artery disease interventions [40].

Furthermore, AI’s fusion with genomics, transcriptomics, and metabolomics is revolutionizing biomarker discovery. Recent studies illustrate AI’s ability to integrate multi-omic datasets, uncovering complex molecular associations that would remain undetectable using traditional statistical methodologies [41,42]. By merging demographic, clinical, imaging, and multi-omics data, AI is ushering in a new era of precision medicine, one that moves beyond population-based guidelines to tailor therapeutic strategies to individual patients.

### 3.4. Preventive Cardiology

Artificial intelligence is redefining preventive cardiology by enhancing traditional risk stratification tools and enabling individualized interventions. Historically, cardiovascular prevention has relied heavily on population-derived risk scores such as the Framingham Risk Score [43,44], which, while useful at a population level, often fail to account for the nuances of individual risk profiles. AI overcomes these limitations by integrating diverse data streams—ranging from electronic health records and imaging to genomics, metabolomics, and social determinants of health—to generate dynamic, personalized risk predictions and outperform conventional scores [45].

An emerging paradigm in this realm is the identification of subclinical disease states or high-risk inflammatory phenotypes. For instance, deep learning applied to coronary computed tomography angiography (CCTA) has enabled automated quantification of pericoronary fat inflammation, which is an early driver of atherosclerotic plaque destabilization [46]. This is particularly relevant in prediabetic patients, where chronic low-grade inflammation is both a hallmark and a prognostic driver of disease progression. In this context, the studies have shown that in certain prediabetic populations, elevated levels of sodium-glucose cotransporter 2 (SGLT2) and leptin, alongside altered sirtuin 6 (SIRT6) expression and microRNA levels, were associated with higher rates of major adverse cardiac events [47]. AI could leverage such molecular and imaging biomarkers to detect inflammatory signatures in at-risk individuals and optimize therapeutic decisions, such as the early administration of metformin and other drugs, in such populations [48]. Another promising avenue is the integration of AI into wearable devices and remote monitoring systems to detect real-time physiologic perturbations indicative of increased cardiovascular risk. Coupled with digital twins or personalized health avatars, these systems could deliver tailored lifestyle, pharmacologic, or behavioral recommendations, thus translating molecular insights into everyday preventive care [49]. Finally, broader efforts like the CardioSight platform from Singapore represent cutting-edge AI-enabled surveillance systems that can visualize cardiovascular risk factor data geospatially and in real time. CardioSight maps parameters like HbA1c and lipid levels, not only to identify individuals at risk, but also expose population-level care gaps [50].

**Table 1 biomedicines-13-01019-t001:** Selection of key studies on advanced AI applications in cardiovascular medicine.

Study	Task	AI Approach	Input Modalities	Outcomes and Explainability *
Agibetov et al., 2021 [51]	Automatic diagnosis of CA using CMR	Pretrained VGG16 CNN (transfer learning)	CMR imaging	AUC: 0.96, SEN: 94%, SPE: 90%; similar performance to expert radiologists
Akita et al., 2024 [29]	Predict myocardial fibrosis from TTE	CNN	TTE, CMR (LGE presence) for ground truth	AUC: 0.86; outperformed reference model based on clinical parameters
Attia et al., 2019 [16]	Screen and predict future LVSD using ECG	CNN (6 layers with Relu activation, batch normalization, max pooling, and spatial fusion)	12-lead ECG	AUC: 0.93, SEN: 86.3%, SPE: 85.7%, ACC: 85.7%; superior AUC compared to traditional screening
Attia et al., 2019 [19]	Identify AF from normal sinus rhythm ECGs	CNN	12-lead ECG	AUC: 0.87, SEN: 79.0%, SPE: 79.5%; outperformed standard risk scores
Bos et al., 2021 [52]	Detect LQTS from 12-lead ECGs	CNNs (10 blocks of convolutional, batch normalization, Relu, and max pooling layers)	12-lead ECG	AUC: 0.90; improved performance over QTc interval measurement-based diagnosis, particularly in concealed cases
Chen et al., 2024 [53]	Classification of Fabry Cardiomyopathy vs. HCM	3D Convolutional Neural Network (3D ResNet18)	CMR imaging	ACC: 91%, AUC: 0.91, F1-score: 0.85; XAI with Grad-CAM
Cohen-Shelly et al., 2021 [54]	Detection of moderate-to-severe AVS from ECG	CNN	ECG	AUC: 0.85, SEN: 78%, SPE: 74%; XAI with saliency maps
DeGroat et al., 2024 [41]	Novel cardiovascular biomarkers using multi-omics data	XGBoost classifier with Bayesian hyperparameter tuning	Multi-omics (RNA-seq, whole-genome sequencing, clinical demographics)	High prediction accuracy for cardiovascular disease (identified biomarkers showed strong literature-based validation); XAI with SHAP
Drouard et al., 2024 [42]	Prediction of cardiovascular risk factors from multi-omic data	Supervised and semi-supervised autoencoders, meta-learners	Blood-derived metabolomics, epigenetics, transcriptomics	AUC improvement of 0.09–0.14 for transcriptomics and 0.07–0.11 for metabolomics with transfer learning; feature importance analysis
Economou Lundeberg et al., 2023 [22]	Predicting the risk of VT using 24-h ambulatory ECG data	Elastic Net Regression Model	Ambulatory ECG	AUC: 0.76, NPV of bottom quintile: 98.2%
Gavidia et al., 2024 [24]	Prediction of AF onset using wearable device data	CNN (479 layers, EfficientNetV2)	R-to-R interval signals from wearable devices	ACC: 83%, F1-score: 85%, Warning time: 30.8 min before AF onset
Goto et al., 2021 [26]	Automated detection of CA using AI-based ECG and TTE models	ECG model: 2D CNN (18 layers); Echo model: 3D-CNN (trained on video sequences of apical 4-chamber views)	ECG and TTE	ECG model: C-statistics of 0.85–0.91 across multiple institutions; Echo model: C-statistics of 0.89–1.00; outperformed expert cardiologists across multiple datasets
Gregoire et al., 2024 [25]	Prediction of short-term AF onset using HRV from Holter ECG recordings	XGBoost decision tree trained on HRV parameters	2-lead Holter ECG data	AUC: 0.92, AUPRC: 0.92, ACC: 84.5%, SEN: 83.0%, SPE: 86.6%, F1-score: 86.2%
Grogan et al., 2021 [27]	Detect CA from a standard 12-lead ECG	Deep Neural Network (AI-enhanced ECG model) trained to predict CA presence	12-lead ECG, single-lead, and 6-lead ECG subsets	AUC: 0.91, SEN: 0.84, SPE: 0.85, PPV: 0.86, NPV: 0.84; predicted CA more than 6 months before clinical diagnosis
Jentzer et al., 2021 [55]	Identification of LVSD using ECGs	CNN trained on nearly 100,000 ECGs	12-lead ECG, TTE for ground truth	AUC: 0.83, SEN: 72.8%, SPE: 77.8%, NPV: 84.9%, ACC: 76.1%
Jiang et al., 2024 [21]	Detection of congenital LQTS and differentiation of genotypes based on 12-lead ECGs	CNN (based on ResNetv2, trained using Bayesian optimization)	12-lead ECG, genetic testing for KCNQ1 and KCNH2 variants	LQTS detection AUC: 0.93, Genotype differentiation AUC: 0.91, SEN: 0.90, F1-score: 0.84; outperformed expert-measured QTc in detecting LQTS, including concealed cases
Ko et al., 2020 [18]	Detection of HCM using a 12-lead ECG	CNN	12-lead ECG	AUC: 0.96, SEN: 87%, SPE: 90%, PPV: 31%, NPV: 99%; outperformed standard ECG interpretation, distinguishing HCM from LVH and normal ECGs
Kolk et al., 2023 [34]	Prediction of non-arrhythmic mortality in patients with primary prevention ICD	XGBoost	Multimodal data: 12-lead ECG and clinical variables (demographics, medical history, medications, lab values)	AUC (internal validation): 0.90, AUC (external validation): 0.79, SEN: 70.3%, SPE: 69.6%, F1-score: 74.6%; outperformed traditional risk scores; XAI with SHAP and ECG heatmaps
Kolk et al., 2024 [32]	Prediction of malignant ventricular arrhythmias in patients with non-ischemic heart failure	DEEP RISK model: Residual VAE, XGBoost	CMR (LGE), 12-lead ECG, clinical data (demographics, medical history, lab values, medications)	AUC: 0.84, SEN: 0.98, SPE: 0.73, AUPRC: 0.31; XAI with SHAP and latent traversal visualizations
Lampert et al., 2023 [56]	Prediction of LVSD in patients with PVCs	Pretrained CNN (ResNet-152)	12-lead ECG	AUC: 0.79 (internal validation), AUC: 0.85 (external validation); outperformed traditional PVC burden-based risk assessment methods; XAI with Grad-CAM
Lee et al., 2016 [57]	Prediction of VT one hour before its occurrence using NNs	NN with one hidden layer (5–13 hidden neurons)	14 HRV and RRV parameters	AUC: 0.93, SEN: 88.2%, SPE: 82.4%, ACC: 85.3%; outperformed traditional VT prediction methods
Lee et al., 2023 [58]	Prediction of significant CAD on coronary CTA in asymptomatic individuals	NN with multi-task learning and feature selection	Coronary CTA, demographics, clinical and laboratory data	AUC: 0.78, ACC: 71.6%; outperformed CAD consortium score (AUC: 0.67), UDF score (AUC: 0.71), and PCE (AUC: 0.72); XAI with SHAP
Libiseller-Egger et al., 2022 [59]	Identification of the genetic basis of cardiovascular age and its impact on cardiovascular risk	CNN	12-lead ECG, GWAS using UK Biobank data	Identified eight genetic loci associated with ECG-derived cardiovascular age, with a correlation between delta age and risk factors; feature importance analysis
Liu et al., 2023 [17]	AI TTE for diagnosing four entities: ASD, DCM, HCM, and prior MI	Pre-trained Inception-V3 for feature extraction from single frames, followed by a multiple-layer 1D CNN for video-level classification	TTE (apical 4-chamber videos)	AUCs: 0.99% (ASD), 0.98% (DCM), 0.99% (HCM), 0.98% (prior MI), and 0.98% (Normal); performed comparably to or better than cardiologists; XAI with CAM
Melo et al., 2023 [60]	Detection of BrS from an ECG without a sodium channel blocker challenge	Feed-forward NN (optimized with ensemble learning)	12-lead ECG (9 leads used for classification: I, II, III, V1–V6)	Training cohort: ACC: 84%, SEN: 85.4%, SPE: 82.4%, AUC: 0.91; Validation cohort: ACC: 88.4%, SEN: 79.6%, SPE: 93.6%, AUC: 0.93; outperformed clinicians in BrS diagnosis; XAI with ECG heatmaps
Morita et al., 2021 [31]	Prediction of the positive genotype in patients with HCM using TTE images	CNN	TTE images (parasternal short- and long-axis, apical 2-, 3-, 4-, and 5-chamber views)	AUC: 0.86 (CNN & Mayo Score) vs. 0.72 (Mayo Score), AUC: 0.84 (CNN & Toronto Score) vs. 0.75 (Toronto Score); improved SEN, SPE, PPV, and NPV; outperformed reference models; XAI with Grad-CAM
O’Driscoll et al., 2023 [61]	Predicting MACE and all-cause mortality from LVEF and GLS	Machine learning-based AI algorithm for automated TTE image processing	AI-contoured left ventricular function from TTE images, GLS, and LVEF	AI-calculated LVEF and GLS were independently associated with MACE
Oikonomou et al., 2024 [33]	AI tool for guiding cardiac testing in stable chest pain patients	Machine-learning-derived algorithm (XGBoost) applied to phenomapping-based decision support	Demographic and clinical data (age, sex, BMI, hypertension, diabetes, lipid panel, medications, others)	AI-aligned testing was associated with lower risk of MI or death; improved CAD detection rates in anatomical-first testing; feature importance analysis
Oikonomou et al., 2024 [62]	Detection and risk stratification of AVS progression using a video-based AI biomarker	Deep learning-based video AI model (DASSi), saliency maps	TTE, CMR	Faster AVS progression prediction, Hazard Ratios for aortic valve replacement: 4.97 for YNHHS cohort, 4.04 for CSMC cohort, 11.38 for UK Biobank; XAI with saliency maps and feature importance analysis
Raghunath et al., 2021 [63]	Predicting new-onset AF from a 12-lead ECG to enable targeted screening	CNN trained on 12-lead ECG signals	12-lead ECG, patient demographics (age, sex)	AUC: 0.85, AUPRC: 0.22 for predicting AF within 1 year; SEN: 69%, SPE: 81%; Hazard Ratio for high- vs. low-risk groups: 7.2 over 30 years; outperformed CHARGE-AF and an XGBoost model using only age and sex
Sahashi et al., 2024 [64]	Deep learning-based estimation of CMR tissue characteristics from TTE	Video-based CNN trained on TTE videos to predict CMR-derived labels	TTE, CMR for ground truth	AUC: 0.56–0.87 for different CMR parameters
Schwartz et al., 2022 [38]	Evaluating the feasibility, accuracy, and safety of AI-based left atrium reconstruction	Model-based FAM using an AI algorithm trained on >300 left atrial CTA	Electroanatomic mapping with CARTO 3, ICE, fluoroscopy, and cardiac CTA	Reduced mapping and fluoroscopy time during PVI; no major complications
Sun et al., 2024 [65]	Detecting CAD from ECG and PCG signals	Parallel CNN network with autoencoder, SVM classifier	ECG, PCG	ACC: 98.45%, SEN: 98.6%, SPE: 98.6%, F1-score: 98. 9%; outperformed various CAD detection methods
Surucu et al., 2021 [66]	Prediction of AF episodes before they occur with HRV	CNN with preprocessing steps	ECG-derived HRV	ACC: 100%, PRE: 100%, SEN: 100%, F1-score: 100%
Tokodi et al., 2023 [67]	Prediction of RVEF using 2D TTE videos	Spatiotemporal CNNs, ensemble model of three best-performing networks, saliency maps	2D TTE videos (apical 4-chamber view)	Mean absolute error for RVEF: 4.57% (internal validation), 5.54% (external validation); ACC for RVEF <45%: 78.4% vs. human expert: 77.0%; performance comparable to human experts, but the model had higher SEN at the cost of SPE; XAI with 2D TTE heatmaps
Torres Soto et al., 2022 [68]	Differentiating HCM from HTN LVH using multimodal AI	Multimodal deep learning model (LVH-Fusion; late-average fusion, late-ranked fusion, and pre-trained and random late fusion models)	ECG, TTE	F1-score: 0.73 (HCM), 0.96 (HTN); SEN: 0.73; SPE: 0.96; outperforms single-modal models; false discovery rate reduced from 0.59 to 0.27; outperformed cardiologists in correctly classifying HCM and HTN; XAI with SHAP
Venkat et al., 2023 [69]	Identification of genes associated with cardiovascular risk using RNA-seq	RF model, recursive feature elimination, SelectKBest feature selection	RNA-seq (gene expression), demographic data (age, gender, race)	ACC for HF: 90.9%, for AF: 95%, for other cardiovascular diseases: 95.9%; feature importance analysis
Yang et al., 2021 [70]	Prediction of ischemia and prognosis from coronary plaque features	Machine learning-based feature selection (Boruta algorithm, hierarchical clustering)	Coronary CTA, FFR and clinical data for ground truth	AUC for low FFR: 0.8, for 5-year vessel-oriented composite outcome: 0.71; improved ischemia prediction; feature importance analysis
Zhang et al., 2022 [71]	Identification of CAD-related genes from gene networks	Graph Convolutional Network, three-layer deep learning model	Gene expression, gene interaction networks	AUC: 0.75, AUPR: 0.78, ACC: 66.9%; better performance than other models
Zhang et al., 2023 [72]	Identification of CA using TTE and machine learning	Various machine models: LR, RF, SVM, others; texture feature extraction	TTE, myocardial texture analysis	AUC: 0.71–0.81, across models; LR had the best performance (F1-score: 0.34, SEN: 0.21, SPE: 1.0); outperformed traditional TTE
Zhao et al., 2022 [73]	Prediction of HF with preserved ejection fraction using DNA methylation and clinical data	Deep learning framework integrating Factorization Machine-based Neural Network, LASSO, and XGBoost-based feature selection	DNA methylation (CpG sites), clinical data	AUC: 0.90, Hosmer–Lemeshow statistic: 6.17 (p: 0.632); outperformed existing risk models
Zhou et al., 2023 [30]	Differentiation of ischemic cardiomyopathy from DCM using TTE	Machine learning models: XGBoost (best-performing), logistic regression, RF, NN	TTE, demographic data	XGBoost: AUC: 0.93, ACC: 75%, SEN: 72%, SPE: 78% (internal validation); AUC: 0.80, ACC: 78%, SEN: 64%, SPE: 93% (external validation)

* Explainability mechanisms are reported only for studies that explicitly incorporate such methods. Abbreviations: ACC, accuracy; AF, atrial fibrillation; AI, artificial intelligence; NN, neural network; ASD, atrial septal defect; AUC, area under the curve; AUPRC, area under the precision–recall curve; AVS, aortic valve stenosis; BMI, body mass index; BrS, Brugada syndrome; CA, cardiac amyloidosis; CAD, Coronary Artery disease; CAM, class activation mapping; CHARGE-AF, cohorts for heart and aging research in genomic epidemiology–atrial fibrillation score; CMR, cardiac magnetic resonance; CNN, convolutional neural network; CTA, computed tomography angiography; DCM, dilated cardiomyopathy; ECG, electrocardiogram; F1-score, F1 Score (harmonic mean of precision and recall); FAM, fast anatomical mapping; FFR, fractional flow reserve; GLS, global longitudinal strain; Grad-CAM, Gradient-weighted Class Activation Mapping; GWAS, genome-wide association study; HCM, hypertrophic cardiomyopathy; HF, heart failure; HRV, heart rate variability; HTN, hypertension; ICD, implantable cardioverter-defibrillator; ICE, intracardiac echocardiography; KCNH2/KCNQ1, potassium voltage-gated channel subfamily h member 2/Q member 1; LGE, late gadolinium enhancement; LR, logistic regression; LQTS, long QT syndrome; LVEF/RVEF, left/right ventricular ejection fraction; LVH, left ventricular hypertrophy; LVSD, left ventricular systolic dysfunction; MACE, major adverse cardiac events; MI, myocardial infarction; NPV, negative predictive value; PCG, phonocardiogram; PRE, precision; PVC, premature ventricular complexes; PVI, pulmonary vein isolation; RF, random forest; RNA, ribonucleic acid; RRV, respiratory rate variability; SEN, sensitivity; SHAP, SHapley Additive exPlanations; SPE, specificity; SVM, support vector machine; TTE, transthoracic echocardiography; VAE, variational autoencoder; VGG16, visual geometry group 16-layer CNN; VT, ventricular tachycardia; XAI, explainable artificial intelligence; XGBoost, extreme gradient boosting.

## 4. Challenges and Future Opportunities

While artificial intelligence is reshaping cardiovascular care, several challenges must be addressed to ensure its effective and ethical integration into clinical practice. These hurdles range from technical limitations to regulatory and ethical concerns. Overcoming them will require interdisciplinary collaboration, continuous technological advancements, and a commitment to transparency and equity in AI applications.

### 4.1. The Black-Box Problem: Enhancing AI Transparency and Explainability

One of the most significant challenges in medicinal AI is its “black-box” nature, where models generate predictions without clear explanations in their decision-making process [9]. DL models, particularly neural network-based ones, rely on complex, multi-layered computations that make it difficult for clinicians to interpret or validate their outputs. This lack of transparency fosters skepticism among medical professionals and hinders clinical adoption.

To mitigate this, the field is increasingly embracing explainable AI (XAI), which focuses on developing models that can provide human-understandable explanations for their predictions. Techniques such as SHapley Additive Explanations (SHAP) and feature importance scoring [74], and Gradient-weighted Class Activation Mapping (Grad-CAM) and saliency maps [75], are now starting to appear, offering insights into what the model focuses on. As detailed in Table 1, some recent paradigms of AI in cardiovascular care have also started to adopt XAI methods, from SHAP values to rank pre-determined features [20,34,41,58], to heatmap visualizations for raw ECG signals and imaging [17,31,53,60]. These approaches provide transparency, improve clinical trust, and support hypothesis generation in biomarker discovery and phenotyping.

However, important limitations remain. For instance, saliency maps often only indicate where the model is attending (e.g., a region of an image or a time series segment), but not necessarily why that focus leads to a given classification [76]. Similarly, methods like SHAP provide statistical insights into feature contributions but may not reflect true causal relationships or biological plausibility [77]. Despite the efforts to integrate XAI with clinical reasoning and provide standardized frameworks for AI development in cardiology [78], the complexity of interpreting multimodal model outputs and challenges in validating XAI-derived insights at the bedside represent ongoing gaps. Therefore, while significant progress has been made in embedding explainability into AI systems—especially in genomics, imaging, and electrophysiology—further advances are needed to ensure transparency and trustworthiness in AI-enhanced clinical decision-making. Future efforts should aim to develop clinically-grounded XAI frameworks, integrate domain-specific ontologies, and explore causality-aware explanations that can bridge the gap between prediction and actionable understanding.

### 4.2. Bias in AI Models: Ensuring Fair and Equitable AI

AI models are only as good as the data they are trained on, and biases present in training datasets can lead to systemic errors in AI-based decision-making [2]. If an AI model is primarily trained on datasets that do not adequately represent diverse populations—including underrepresented racial and ethnic groups, gender minorities, or individuals with rare conditions—it may produce biased and unreliable predictions when applied in broader clinical settings or unseen data.

The risks of biased AI extend beyond incorrect diagnoses; they can also exacerbate existing healthcare disparities. For example, if an AI system is primarily trained on ECG data from male patients, it may perform less accurately for female patients, leading to misdiagnoses or suboptimal treatment recommendations. Healthcare providers and AI developers must collaborate to ensure equitable AI-driven healthcare by incorporating diverse, representative datasets and implementing bias mitigation strategies.

### 4.3. Regulatory and Ethical Considerations

The deployment of AI in medicine exists at the intersection of healthcare, technology, regulation, and ethics. While AI-powered tools hold immense potential for improving diagnosis and treatment, their regulation remains a challenge. Regulatory bodies such as the Food and Drug Administration (FDA) [79], European Medicines Agency (EMA) [80], and Medicines and Healthcare products Regulatory Agency (MHRA) [81], are continually developing guidelines for AI-driven medical devices and services, but the rapid pace of AI innovation often outstrips regulatory frameworks.

Additionally, AI introduces ethical dilemmas concerning autonomy, liability, and informed consent. When an AI-driven decision results in patient harm, who bears responsibility: the clinician using the tool, the developers designing the algorithm, or the institution deploying it? Furthermore, how can patients give informed consent for AI-driven decisions when even experts struggle to fully understand how these models function? Establishing clear legal frameworks and ethical guidelines will be essential to integrating AI safely into patient care.

### 4.4. Interdisciplinary Collaboration: Bridging the Gap

AI in cardiovascular medicine sits at the intersection of multiple fields, requiring close collaboration across multiple disciplines, including clinicians, data scientists, biomedical engineers, and regulatory experts. One of the primary barriers to AI adoption is the disconnect between these fields; many clinicians lack the technical expertise to understand and trust AI models, while AI developers often have limited insight into clinical workflows and real-world patient care.

Bridging this gap will require interdisciplinary education and training programs that promote shared knowledge and communication. AI-driven research initiatives should include multidisciplinary teams, ensuring that models are clinically relevant, ethically sound, and aligned with practical medical needs.

### 4.5. Technological Advances and Open Science

Recent advancements in computational power and data infrastructure have significantly improved AI’s ability to process complex cardiovascular data. Some recent breakthroughs include:

High-performance computational hardware: advances in graphic processing units (GPU) and tensor processing units (TPU) enable AI models to analyze continuous ECG recordings, high-resolution imaging, and multimodal data integration in real-time, drastically improving efficiency in AI workflows [82].Self-supervised learning: This method is an emerging paradigm that enables AI models to learn useful representations from unlabeled data by creating surrogate tasks, such as predicting missing parts of ECG signals or reconstructing corrupted imaging inputs. This is particularly valuable in cardiovascular medicine, where labeled datasets are limited and costly to obtain. Self-supervised learning has already shown promising results in certain areas of cardiovascular research, such as echocardiographic view classification [83,84] and anomaly detection in ECGs [85].Federated learning: Federated learning allows multiple institutions to collaboratively train AI models without sharing raw patient data. Instead, model updates are exchanged, preserving patient privacy and complying with data protection regulations [86]. This decentralized approach facilitates access to diverse and representative datasets, improving model generalizability and robustness. Federated learning has already marked some use cases in the field, such as distributed ECG classification [87], cross-center imaging tasks [88], and model development for genomic research [89].Foundational models in cardiovascular AI: Foundational models—large-scale, pre-trained neural networks initially developed for broad tasks—are increasingly adapted for cardiovascular medicine [90,91]. These models, such as HeartBEiT for ECG interpretation and EchoCLIP for echo-imaging analysis, are trained on vast and diverse datasets, can be fine-tuned for a variety of downstream clinical applications [92,93,94]. Similar to ChatGPT (OpenAI) in natural language processing, foundational models in medicine can capture universal patterns across different data modalities and patient populations, and offer enhanced performance, adaptability, and transferability to novel cardiovascular tasks, even in low-resource or data-scarce environments.

Despite these technological strides, data and code accessibility remain major barriers to AI-driven cardiology [2]. Many state-of-the-art AI models are developed within academic and private-sector silos, with limited transparency regarding their training and validation processes. This lack of open access hinders reproducibility, making it difficult for the broader scientific community to verify results and build upon existing advancements.

To address this, data-sharing initiatives and open-source AI models should be prioritized. Large-scale cardiovascular datasets, such as the UK Biobank [95] and the Multi-Ethnic Study of Atherosclerosis (MESA) [96], along with AI-driven repositories like PhysioNet [97], can facilitate collaborative research and innovation. Governments, research institutions, and industry leaders must balance data privacy concerns with the need for open AI development. Additionally, standardized evaluation benchmarks should be established to ensure that AI models are rigorously tested across diverse patient populations before clinical deployment. By fostering an open-science approach, AI advancements can be accelerated while maintaining transparency, fairness, and clinical trust.

## 5. Conclusions

Artificial intelligence is transforming cardiovascular medicine, not by replacing clinicians, but by augmenting their capabilities. AI should not be limited to automating existing workflows or mimicking human decision-making. From ECG interpretation to multi-omics integration for diagnostics, prognostics, and preventive medicine, AI excels where human cognition meets its limits: in synthesizing multimodal data and integrating diverse information sources to drive the next generation of precision medicine.

Challenges still persist since many models remain opaque, raising concerns around explainability, bias, and clinical trust. While recent advances in AI, such as XAI, federated learning, and foundational models, show promise, broader adoption will require transparent validation, open and diverse datasets, and stronger interdisciplinary collaboration.

The future of cardiovascular AI lies not in mimicking human decision-making but in extending it by empowering physicians with insights beyond human cognitive reach. *We do not expect a bird to swim or a fish to fly*; similarly, we should not request from AI to think like a human but rather to do what clinicians cannot. Embracing this synergy will redefine the frontiers of precision cardiovascular care.

## Figures and Tables

**Figure 1 biomedicines-13-01019-f001:**
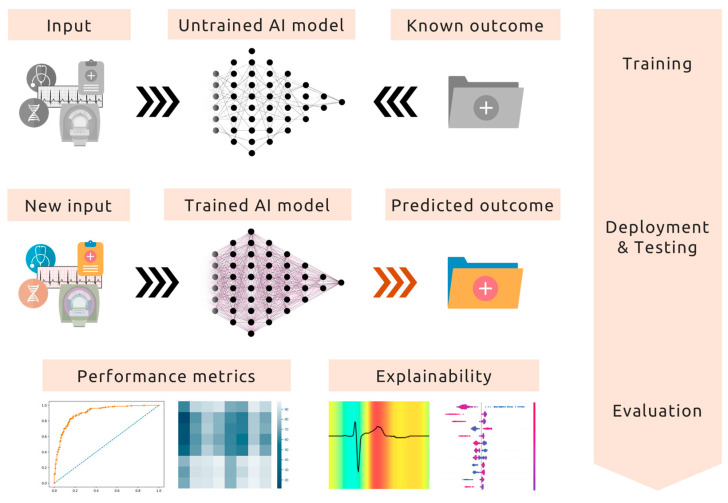
Pipeline of artificial intelligence (AI) model development in cardiovascular medicine.

**Figure 2 biomedicines-13-01019-f002:**
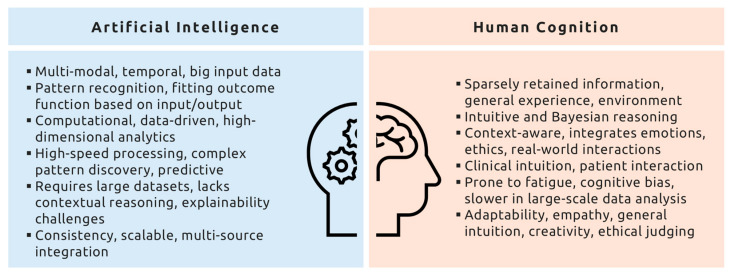
Conceptual comparison of artificial intelligence and human cognition traits in clinical analytical thinking.

## Data Availability

No new data were created.

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
