# Peer review of "Hearts, Data, and Artificial Intelligence Wizardry: From Imitation to Innovation in Cardiovascular Care"

_biomedicines, 2025, doi:10.3390/biomedicines13051019_

Round 1

Reviewer 1 Report

Comments and Suggestions for Authors

Unfortunately, the study does not contain the elements that a review should have. For example, from which sources was the literature reviewed? Which keywords were used during the search? How were the results then filtered? 

In addition, the sections of the study without references are more repetitions of known information than interesting findings. For example, in section 4.1, it is stated that the focus should be on XAI, but there is already a lot of work being done in this area. 

It is also a question mark as to how this review and the results were obtained, as the systematic study mentioned at the beginning is not mentioned.

Finally, the Conclusions section is very weak and should be expanded.

Author Response

  1. Comment: Unfortunately, the study does not contain the elements that a review should have. For example, from which sources was the literature reviewed? Which keywords were used during the search? How were the results then filtered?
    Answer: Thank you for this important observation. This article is a narrative review, not a systematic review, in line with the formats accepted by Biomedicines (www.mdpi.com/journal/biomedicines/instructions). Nonetheless, to enhance transparency and reproducibility, we now provide a new subsection (2.1 “Literature Search Strategy”), where we outline our approach, including databases, search terms, and inclusion/exclusion criteria (lines: [88–103]).
  2. Comment: The sections of the study without references are more repetitions of known information than interesting findings. For example, in section 4.1, it is stated that the focus should be on XAI, but there is already a lot of work being done in this area.
    Answer: We appreciate this feedback. Section 4.1 has been thoroughly revised to include recent references and acknowledge the expanding literature on Explainable AI (XAI). Additionally, we updated Table 1 to include notes on the XAI techniques employed in the referenced studies. We also highlight existing limitations—for instance, the restrcition imposed by saliency maps to reveal not only where the model focuses, but also why. We now better contextualize our discussion by clearly stating both the progress made and the remaining gaps that require further investigation (Table 1 and lines: [286–309]).
  3. Comment: It is also a question mark as to how this review and the results were obtained, as the systematic study mentioned at the beginning is not mentioned.
    Answer: Thank you for pointing this out. Although this is a narrative review, as not necessarily adopting a pre-defined framework for systematically summarizing evidence, we added a dedicated section on methodology (see response to Comment 1) to avoid confusion and enhance the manuscript’s clarity and rigor.

  4. Comment: Finally, the Conclusions section is very weak and should be expanded.
    Answer: We fully agree and thank you for this suggestion. The Conclusions section has been significantly expanded to summarize key messages, open challenges, and practical implications, and present a forward-looking vision into the role of advanced AI in cardiovascular medicine. (lines: [397–411]).

Reviewer 2 Report

Comments and Suggestions for Authors

The paper is titled as 'Hearts, Data, and Artificial Intelligence Wizardry: From Imitation to Innovation in Cardiovascular Care' and it is aimed to explore the evolving role of artificial intelligence (AI) in cardiovascular medicine, highlighting its applications in diagnostics, risk assessment, and precision medicine while addressing challenges related to AI's transparency, bias, and integration into clinical practice. 

The topic is interesting and one of the hot topics in the field of cardiovascular medicine and AI-driven healthcare, and it fits in the scope of Biomedicines Journal.

Firstly;
"Figure 1. Pipeline of artificial intelligence (AI) model development in cardiovascular medicine" is very expressive to draw the aim of the AI.
Additionally in the ongoing paper the authors made a comparison according to performance metrics and used ML algorithms.
They also emphasize the use of XAI here, However, not mentioned the concept in a comparative table.
I want to see its addition to Table 1. Selection of key studies on advanced AI applications in cardiovascular medicine

Secondly;
I want to learn how the compared paper are selected from the Literature?
What is your approach for this?
In Which dataatabase you made your sech?
What is you inclusion/exclusion criteria?

Thirdly;
The paper is generally focussed on predicting cardiovascular diseases at an early stage before symptoms appear.
However, AI applications in preventive medicine, such as personalized lifestyle recommendations based on AI-analyzed risk factors is also an hot topic.

I want to see a subsection about this.

Finally,
In the conclusion section,
Is the following sentence a conclusion from the paper?
"We cannot push a bird to swim or a fish to fly; instead, we must embrace their unique capabilities and explore the possibilities they offer."

Author Response

  1. Comment: Figure 1. Pipeline of artificial intelligence (AI) model development in cardiovascular medicine" is very expressive to draw the aim of the AI. Additionally in the ongoing paper the authors made a comparison according to performance metrics and used ML algorithms. They also emphasize the use of XAI here, However, not mentioned the concept in a comparative table. I want to see its addition to Table 1. Selection of key studies on advanced AI applications in cardiovascular medicine.
    Answer: Thank you very much for this excellent suggestion. We have revised Table 1, renaming the final column (previously "Outcome") to “Outcomes and Explainability”. For each study, we now indicate whether Explainable AI techniques were used (e.g., saliency maps, SHAP values). This addition strengthens the clarity and utility of the comparative analysis (Table 1]).

  2. Comment: How were the compared papers selected from the literature? What was the approach? Which database was used? What were the inclusion/exclusion criteria?
    Answer: Thank you for your question. While our article follows a narrative rather than a systematic review format, in line with the accepted articles types by the journal, we have now included a dedicated subsection (2.1) titled “Literature Search Strategy”. This explains our data sources, search keywords, and inclusion/exclusion criteria, improving the transparency of our selection process (lines: [88–103]).

  3. Comment: The paper is generally focussed on predicting cardiovascular diseases at an early stage before symptoms appear. However, AI applications in preventive medicine, such as personalized lifestyle recommendations based on AI-analyzed risk factors is also an hot topic. I want to see a subsection about this.
    Answer: Thank you for this valid point. We have now added a dedicated subsection 3.4 “Preventive Cardiology”, where we discuss relevant notions and applications (lines: [220–249]). Key recent studies are cited to support these emerging developments.

  4. Comment: In the conclusion section, Is the following sentence a conclusion from the paper? “We cannot push a bird to swim or a fish to fly; instead, we must embrace their unique capabilities and explore the possibilities they offer.”
    Answer: Thank you for the feedback. We agree that while the sentence is metaphorical, it serves as a summary metaphor for the complementary roles of AI and human clinicians. In the revised version, we have rephrased and supplemented this sentence with a more analytical conclusion, ensuring the final paragraph captures the paper's evidence-based takeaways while retaining the spirit of the metaphor (lines: [407–411]).

Reviewer 3 Report

Comments and Suggestions for Authors
  1. Introduction, lines 61-3, “… the transition from traditional machine learning (ML) techniques to deep learning (DL) models. These models have significantly enhanced diagnostic precision, prognosis assessment, treatment guidance, and risk stratification in cardiovascular care [1]”: in the first paragraph of your manuscript (and in the title), you announce to deal with “cardiovascular care”, but then throughout the text you focus only on cardiological issues. You need at least to add the following appropriate and recent MDPI reference, that not only well explain traditional machine learning techniques and deep learning models, but also shows that AI is involved in other fields of “cardiovascular care”, other than Cardiology (Martelli, E.; Capoccia, L.; Di Francesco, M.; Cavallo, E.; Pezzulla, M.G.; Giudice, G.; Bauleo, A.; Coppola, G.; Panagrosso, M. Current Applications and Future Perspectives of Artificial and Biomimetic Intelligence in Vascular Surgery and Peripheral Artery Disease. Biomimetics (Basel) 2024, 9, 465-477).

  1. Challenges and Future Opportunities, 4.3. Regulatory and Ethical Considerations, line 256, “… FDA [63], EMA [64], and MHRA [65] …”: each acronym must be explained the first time it is used.

Abbreviations: you can delete this last section (so saving save space for the publisher), since its explanation of each acronym has been already performed throughout the text.

Author Response

  1. Comment: 1. Introduction, lines 61-3, “… the transition from traditional machine learning (ML) techniques to deep learning (DL) models. These models have significantly enhanced diagnostic precision, prognosis assessment, treatment guidance, and risk stratification in cardiovascular care [1]”: in the first paragraph of your manuscript (and in the title), you announce to deal with “cardiovascular care”, but then throughout the text you focus only on cardiological issues. You need at least to add the following appropriate and recent MDPI reference, that not only well explain traditional machine learning techniques and deep learning models, but also shows that AI is involved in other fields of “cardiovascular care”, other than Cardiology (Martelli, E.; Capoccia, L.; Di Francesco, M.; Cavallo, E.; Pezzulla, M.G.; Giudice, G.; Bauleo, A.; Coppola, G.; Panagrosso, M. Current Applications and Future Perspectives of Artificial and Biomimetic Intelligence in Vascular Surgery and Peripheral Artery Disease. Biomimetics (Basel) 2024, 9, 465-477).
    Answer: Thank you for your insightful suggestion. We have now revised the Introduction and Section 2 to acknowledge the broader scope of cardiovascular care, including vascular surgery and peripheral artery disease, and we have included the reference to Martelli et al., Biomimetics (Basel) 2024, 9, 465–477 to appropriately reflect this breadth (lines: [64, 74, 207–212]).

  2. Comment: 4. Challenges and Future Opportunities, 4.3. Regulatory and Ethical Considerations, line 256, “… FDA [63], EMA [64], and MHRA [65] …”: each acronym must be explained the first time it is used.
    Answer: Thanks for pointing this out. This has been corrected (lines: [327–328]).

  3. Comment: Abbreviations: you can delete this last section (so saving save space for the publisher), since its explanation of each acronym has been already performed throughout the text.
    Answer: Thank you for this feedback. We have removed the Abbreviations section, and definitions are now provided only in-text, at their first appearance.

Round 2

Reviewer 1 Report

Comments and Suggestions for Authors

Although the authors have responded to my criticisms and made minor corrections, I believe that the responses and corrections are insufficient. Therefore, I believe that it is not appropriate to accept the study.

Author Response

Comment: Although the authors have responded to my criticisms and made minor corrections, I believe that the responses and corrections are insufficient. Therefore, I believe that it is not appropriate to accept the study.

Answer: Thank you for your continued assessment. While we understand that no specific new revision requests were made, we have nonetheless implemented the following additional changes to strengthen the scientific content, organization, and impact of the manuscript:

  • Improved flow and precision in Section 3.3: Slightly restructured the paragraph discussing AI in procedural planning to ensure even clearer delineation between cardiology and vascular applications. Incorporated a clearer lead-in sentence: “AI is now increasingly integrated into interventional workflows, guiding precision-driven procedural planning across both cardiology and vascular domains."
  • Expanded Table 1 captions: Clarified that explainability mechanisms are reported only for studies that explicitly incorporate or describe such methods.
  • Clarified scope in the Introduction: Minor rewording to reinforce that cardiovascular care spans both cardiac and vascular disciplines.

Reviewer 2 Report

Comments and Suggestions for Authors

The authors have adequately addressed the concerns raised in the previous round of reviews and made the necessary corrections accordingly. The revisions improve the clarity and quality of the manuscript. Based on the current version, I believe the paper meets the required standards and can be accepted as is.

Author Response

Comment: The authors have adequately addressed the concerns raised in the previous round of reviews and made the necessary corrections accordingly. The revisions improve the clarity and quality of the manuscript. Based on the current version, I believe the paper meets the required standards and can be accepted as is.

Answer: We are sincerely grateful for your supportive feedback. We appreciate your recognition of the improvements and your recommendation for acceptance. Thank you again for your valuable contribution throughout the review process.

Round 3

Reviewer 1 Report

Comments and Suggestions for Authors

I think it is appropriate to accept the study in this form.